# Non-Invasive Brain Stimulation Effects on Biomarkers of Tryptophan Metabolism: A Scoping Review and Meta-Analysis

**DOI:** 10.3390/ijms23179692

**Published:** 2022-08-26

**Authors:** Cristian G. Giron, Tim T. Z. Lin, Rebecca L. D. Kan, Bella B. B. Zhang, Suk Yu Yau, Georg S. Kranz

**Affiliations:** 1Department of Rehabilitation Sciences, The Hong Kong Polytechnic University, Hong Kong SAR, China; 2Mental Health Research Center (MHRC), The Hong Kong Polytechnic University, Hong Kong SAR, China; 3Department of Psychiatry and Psychotherapy, Comprehensive Center for Clinical Neurosciences and Mental Health, Medical University of Vienna, 1090 Vienna, Austria; 4The State Key Laboratory of Brain and Cognitive Sciences, The University of Hong Kong, Hong Kong SAR, China

**Keywords:** biomarker, tryptophan, kynurenine, serotonin, electroconvulsive therapy, repetitive transcranial magnetic stimulation, transcranial direct current stimulation

## Abstract

Abnormal activation of the kynurenine and serotonin pathways of tryptophan metabolism is linked to a host of neuropsychiatric disorders. Concurrently, noninvasive brain stimulation (NIBS) techniques demonstrate high therapeutic efficacy across neuropsychiatric disorders, with indications for modulated neuroplasticity underlying such effects. We therefore conducted a scoping review with meta-analysis of eligible studies, conforming with the PRISMA statement, by searching the PubMed and Web of Science databases for clinical and preclinical studies that report the effects of NIBS on biomarkers of tryptophan metabolism. NIBS techniques reviewed were electroconvulsive therapy (ECT), transcranial magnetic stimulation (TMS), and transcranial direct current stimulation (tDCS). Of the 564 search results, 65 studies were included with publications dating back to 1971 until 2022. The Robust Bayesian Meta-Analysis on clinical studies and qualitative analysis identified general null effects by NIBS on biomarkers of tryptophan metabolism, but moderate evidence for TMS effects on elevating serum serotonin levels. We cannot interpret this as evidence for or against the effects of NIBS on these biomarkers, as there exists several confounding methodological differences in this literature. Future controlled studies are needed to elucidate the effects of NIBS on biomarkers of tryptophan metabolism, an under-investigated question with substantial implications to clinical research and practice.

## 1. Introduction

### 1.1. Rationale

The products of tryptophan (TRP) metabolism have vital physiological roles. These bioactive metabolites include serotonin (5-HT), melatonin, and the kynurenines, all with diverse regulatory functions. Although TRP itself is an essential amino acid, with its availability dependent on dietary intake, only a small amount is used for protein synthesis while most of it is degraded via the 5-HT or kynurenine pathways (Figure 1). The biomarkers of interest in the present review, whose concentrations in various media inform about the activation of TRP metabolism are: TRP, kynurenine (KYN), formylkynurenine, kynurenic acid (KA), quinolinic acid (QA), NAD+, 3-hydroxykynurenine (3-HK), xanthurenic acid (XA), picolinic acid (PA), anthranilic acid (AA), 5-HT, oxitriptan, 5-hydroxytryptamine (5-HT), 5-hydroxyindoleacetic acid (5-HIAA), N-acetylserotonin, and melatonin. 

Healthy intake of dietary TRP and production of these metabolic products are linked to cardiovascular health [1], gut-brain homeostasis [2], cognitive and mood regulation [3,4], and immune and inflammation mediation [5]. To reach the central nervous system (CNS), a portion of dietary TRP is absorbed in the small intestine to be released into the blood stream. It then traverses through the bloodstream and is subsequently transported across the blood–brain barrier by the L-type amino acid transporter, which is also expressed by nerve cells within the CNS. This dietary TRP is a vital substrate for 5-HT synthesis by serotonergic neurons in the CNS, which project from the rostral raphe nuclei to the cerebral cortex. These cortices ubiquitously express critical proteins for 5-HT neurotransmission, such as the highly studied receptors 5-HT_1A_ and 5HT_2A_, and monoamine oxidase-A (MAO-A) [6], among several more receptor subtypes and transporters. Serotonergic neurons also project to limbic and sub-cortical structures, other brain stem regions and the spinal cord—providing an anatomical link between the role of serotonin in behavioral, emotional, cognitive, and motor functions mediated by the nervous system. Activation of the kynurenine pathway generates molecules that are neuroactive and mediate inflammatory response [5,7]. Several highly prevalent neuropsychiatric disorders, such as major depressive disorder (MDD), bipolar disorder, and schizophrenia, present with biomarker concentrations that indicate abnormal activation of these pathways [8,9,10]. Further, there is evidence that the restoration of these abnormal activations coincides with therapeutic effects. For instance, antidepressant effects following anti-inflammatory medication are observed in patients with inflammatory conditions [11] and with depressive disorders [12]. Further, the antidepressant effects of the selective serotonin reuptake inhibitor (SSRI), escitalopram, have been observed to negatively correlate with kynurenic acid (KA) plasma levels, with lower levels predicting greater therapeutic outcomes in patients diagnosed with MDD [13]. In a meta-analysis examining peripheral levels of kynurenines in psychiatric patients compared to healthy controls, Marx et al. [9] observed significantly lower TRP levels in patients with MDD and schizophrenia compared to healthy controls, in addition to elevated kynurenine (KYN) to TRP ratios. This latter observation suggests that the diminished TRP availability may be due to over-activation of the kynurenine pathway. Moreover, compared to healthy controls, KA levels were lower in MDD patients but schizophrenia patients had KA levels comparable to controls [9]. Such unique patterns are proposed to underlie the specific syndromes of these disorders [7], supporting efforts to target these abnormal pathway activations for intervention or to inform treatment selection [5,7]. In further support of these efforts, Haroon et al. [14] assessed the association between blood plasma and cerebrospinal fluid (CSF) biomarkers of immune and kynurenine pathway activation, finding elevations in both mediums in depressed patients compared to healthy controls; these elevations also corresponded to more severe motivation and anhedonia symptoms. As elevated kynurenine pathway activation catabolizes more available TRP, serotonin synthesis is consequently diminished due to lack of substrate. Such findings link over-activation of the kynurenine pathway to psychiatric symptoms that are associated with diminished serotonin activity, such as impaired neuroplasticity [15] and disrupted “re-learning” of appropriate emotion processing in depression [4]. Additionally, disrupted serotonin synthesis following kynurenine pathway over-activation lends further credence to the association between psychiatric symptoms and seemingly distant events that alter TRP degradation toward the kynurenine pathway, such as chronic stress [16] and inflammation [7]. As the authors above, Haroon et al. [14] envisioned using biomarkers of TRP metabolism to guide psychiatric treatment.

Noninvasive brain stimulation (NIBS) techniques have demonstrated therapeutic utility across neuropsychiatric disorders, with varied levels of efficacy and quality of evidence [17]. Those techniques under review here are repetitive transcranial magnetic stimulation (rTMS) [17,18,19,20], transcranial direct current stimulation (tDCS) [17,18], and electroconvulsive therapy (ECT) [17]—each considered non-invasive as no breaching or implantation occurs during treatment [21]. Current efforts are aimed at developing precise protocols for these techniques [22]. However, while elucidating the mechanisms of action of NIBS is critical toward these aims, current theories need to be broadened or supported by molecular evidence to account for the enduring therapeutic effects of brain stimulation [23].

In light of theories predicting a link between improved clinical outcomes and recovered pathway activation, and the need to specify molecular mechanisms of NIBS to optimize protocol selection, efforts to investigate the influence of NIBS on tryptophan metabolism are re-emerging [24]. While there exists reviews relevant to this research question [24,25,26], there is a current need for systematic reviews which assess the effects of NIBS on specific biomarkers of tryptophan metabolism in humans and animal models. This examination is critical to understanding the therapeutic mechanisms of brain stimulation across clinical diagnoses and informing efforts toward precision medicine [22].

### 1.2. Objective

This scoping review systematically surveys the current literature investigating the effects of therapeutic NIBS on biomarkers of tryptophan metabolism and synthesizes findings by qualitative- and meta-analyses. Our findings are discussed in light of current etiological theories of neuropsychiatric disorders and whether they are supported by the observed therapeutic and biomarker effects of NIBS. Our search criteria and research question are summarized in Table 1. Briefly, we sought published studies utilizing therapeutic NIBS, and which probed levels of kynurenine or serotonin pathway metabolites in healthy or diagnosed humans and animal models. No further restrictions were considered.

## 2. Methods

This invited review was designed to conform with the Preferred Reporting Items for Systematic reviews and Meta-Analyses extension for Scoping Reviews (PRISMA-ScR) [27].

### 2.1. Eligibility Criteria

Inclusion criteria were (1) treatment with ECT, rTMS, or tDCS; (2) participants were human and non-human species; (3) levels of the following biomarkers were assessed using any form of biological sample collection from NIBS-treated participants: tryptophan (TRP), kynurenine (KYN), formylkynurenine, quinolinic acid (QA), NAD+, 3-hydroxykynurenine (3-HK), xanthurenic acid (XA), picolinic acid (PA), anthranilic acid (AA), serotonin (5-HT), oxitriptan, 5-hydroxytryptamine (5-HT), 5-hydroxyindoleacetic acid (5-HIAA), N-acetyl-5-HT, and melatonin; 4) effects of NIBS on biomarkers of interest levels was assessed by comparing baseline with concurrent or post-NIBS levels, without restriction to timepoints of sampling. Exclusion criteria were (1) no new findings were published or data was previously reported. Otherwise, there were no further restrictions on experimental design, the language the manuscript was written-in, NIBS parameters or targets, participant characteristics such as age or diagnosis, nor restrictions on sampling method were considered. The rationale for these relaxed restrictions was to maximize the number of included studies for this survey, as we anticipated this literature to be sparse.

### 2.2. Information Sources

We searched the NCBI PubMed and Web of Science databases for studies published from inception until 15 July 2022. Included studies and relevant reviews in our search results were also screened for relevant studies.

### 2.3. Search Strategy

Search terms for NIBS were initially based on techniques in a recent expert review [22], however we focused our search here to ECT, rTMS, and tDCS because preliminary searches to test the feasibility of this review could not identify relevant studies for other NIBS techniques and these techniques are frequently discussed in reviews on the therapeutic efficacy of NIBS and mechanisms of action, e.g, [17]. The subset of TRP metabolites, and corresponding terms for the products of the serotonin and kynurenine pathways that we reviewed were based on recent expert reviews [5,7]. The search queries used for the PubMed and Web of Science search are shown in Appendix A.

### 2.4. Selection and Data Collection

Two independent reviewers (C.G.G. and T.T.Z.L.) conducted the search and initial screening by title and abstract. Both reviewers independently retrieved the identified potential studies, conducted full-article screening, and then extracted data for review. These results were then merged, with any disagreements settled through discussion with G.S.K. No automation tools were used for screening or data extraction.

### 2.5. Data Items

A customized form was used to report relevant data from our included studies: study design; NIBS protocol parameters (e.g., stimulation site, session count); participant characteristics (e.g., species, health condition, age); control group characteristics when available; biomarkers of TRP metabolism including metabolite concentration levels, ratios, sampling source, and time points; and significant and non-significant effects of NIBS on biomarkers and direction of effect (Table 1). These significant effects were retrieved from one-group pretest-posttest designs or group differences in parallel or crossover designs. For studies using a parallel design, we preferred to report results based on group*time interaction over group differences in posttests.

For quantitative analysis, we extracted numerical results necessary for the calculation of the standardized mean difference (SMD). If only statistical graphs were provided, we extracted the values using WebPlotDigitizer (https://automeris.io/WebPlotDigitizer/, accessed on 15 July 2022). Clinical measures of human participants (e.g., effects on clinical assessment scales) were also extracted. If there were multiple clinical measures, we sought the primary outcome in the study.

### 2.6. Qualitative Synthesis Methods

Results from clinical and preclinical studies were visualized as bar charts where the counts for significant changes, in either direction, and nonsignificant changes were stacked to present an overview of the effects of NIBS and biomarkers of TRP metabolism. In this synthesis, we stacked studies assessing the same metabolite and NIBS technique (i.e., ECT, rTMS, or tDCS). That is, we differentiated clinical and preclinical studies and NIBS, but did not perform subgroup analyses of all ways in which these studies vary, for example, dissimilar NIBS parameters. Instead, when patterns suggesting heterogeneity or true effects emerged, we agreed to follow them up by examining the characteristics of the relevant studies.

Studies may report findings from multiple experimental groups, for instance, effects on various health conditions. In these cases, we counted experimental groups instead of the study count towards synthesis. Furthermore, within experimental groups, there may be multiple sampling time points. We determined that the time points associated with significant changes would represent the groups, presuming that those time points were most sensitive to effects by NIBS. Experimental groups with significant changes in opposite directions at different time points would be excluded from the bar charts and followed up specifically. The bar charts and other similar graphs were created using the Python library on Plotly (https://plotly.com/python/, accessed on 15 July 2022).

### 2.7. Quantitative Synthesis Methods by Bayesian Meta-Analysis

Only clinical studies with calculable SMDs underwent meta-analysis. We used the Robust Bayesian Meta-Analysis (RoBMA) method, which applied to the data variety of models resting on different assumptions about effects, heterogeneity, and publication bias [28]. These models were weighted according to their predictability of the data and averaged to draw final inferences. The Bayesian framework allows quantification of evidence for null findings, while more traditional, frequentist approaches cannot distinguish support for null findings from the absence of evidence (in the case of *p* > 0.05). The qualitative analysis of significant and nonsignificant effects, as described above, and classical meta-analysis methods are examples of frequentist approaches. Besides the above advantage given by the framework, RoBMA benefits from its model-averaging feature such that it does not require all-or-none decisions about publication bias and works well under high heterogeneity [28], well-suited for this scoping review. The resulting inferences were presented as inclusion Bayes factors (BF10), representing the strength of evidence for the presence of the meta-analytic item relative to the absence. For example, a BF10 of effects equal to three means the data are three times more likely to have occurred if effects exist, or three times less likely when BF10 equals 1/3. Such a BF10 larger than three or less than 1/3 is tentatively interpreted as moderate evidence supporting or against the presence of the meta-analytic item, respectively, as per [29]. Additionally, >10 or <1/10 indicates strong evidence; >30 or <1/30 indicates very strong evidence; >100 or <1/100 indicates extreme evidence. As we expected a substantial number of clinical studies that used a one-group pretest-posttest or crossover design, we decided not to exclude them from this meta-analysis and calculate the SMD as the change or difference divided by the pretest or control standard deviation (SD). Computing the standard error of such SMDs requires the pretest-posttest correlation (r), and as few studies reported this, we adopted Rosenthal’s estimate of r = 0.7 [30], and used r = 0.5 and 0.9 for sensitivity testing. For studies with a parallel design, we preferred to calculate the SMD as the group difference in posttests divided by the pooled SD. Similar to the approach in our qualitative analysis above, when there were multiple SMD values within an experimental group, for example, due to multiple sampling time points, we chose the one with the largest absolute value, assuming it is most sensitive to the change.

Statistics measures other than the mean and SD were converted to them as per the Cochrane Handbook [31], except the median and first and third quartile were converted according to [32]. This quantitative analysis was executed using JASP version 0.16.2 (https://jasp-stats.org/, accessed on 15 July 2022).

## 3. Results

### 3.1. Selection of Sources of Evidence

A total of 65 studies (Table 2 and Table 3) [33,34,35,36,37,38,39,40,41,42,43,44,45,46,47,48,49,50,51,52,53,54,55,56,57,58,59,60,61,62,63,64,65,66,67,68,69,70,71,72,73,74,75,76,77,78,79,80,81,82,83,84,85,86,87,88,89,90,91,92,93,94,95,96,97] were included after screening 307 records in PubMed and 257 records in Web of Science, with seven of our included studies identified in the references of studies included from databases. A total of 29 studies could not be retrieved or were excluded for not meeting inclusion criteria during full-text screening (Appendix A) [36,98,99,100,101,102,103,104,105,106,107,108,109,110,111,112,113,114,115,116,117,118,119,120,121,122,123,124,125]. Further details of our screening results are shown in the PRISMA flow diagram (Figure 2). All the included studies were published in English.

### 3.2. Characteristics of Included Studiess

38 included studies recruited humans (Table 2) [33,34,35,36,37,38,39,40,41,42,43,44,45,46,47,48,49,50,51,52,53,54,55,56,57,58,59,60,61,62,63,64,65,66,67,68,69,70] and 27 studied animal models (Table 3) [71,72,73,74,75,76,77,78,79,80,81,82,83,84,85,86,87,88,89,90,91,92,93,94,95,96,97]. Clinical assessment outcomes for human participants are available in the Appendix A.

A histogram of the study count versus publication year (Figure 3) shows that ECT research, both clinical and preclinical, dominated the 1970s and 1980s. Around the year 2000, there was an emergence of preclinical TMS studies followed by clinical research. In the past decade, there have been no preclinical ECT studies, while the number of clinical ECT and TMS studies have been growing, in addition to the appearance of tDCS studies. Of the 37 clinical studies (excluding case reports [49]), 25 (68%) used ECT, and 11 (30%) used TMS. A lower percentage of preclinical studies (12 of 27, 44%) employed ECT, while more (14, 52%) used TMS. An overview of the effects of NIBS on the concentrations of each biomarker of interest are shown Figure 4.

The clinical and preclinical research on the effects of NIBS on biomarkers of TRP metabolism show overlapping and distinct interests in terms of studied metabolites and sampling sources (Figure 5). Overall, the most frequently investigated were TRP, 5-HT and 5-HIAA. In clinical studies, metabolites in the kynurenine pathway, notably KYN and KA, were frequently studied, but missing in preclinical studies.

Clinical studies generally lacked randomized control designs and included pharmacotherapy-as-usual. Of the 37 clinical studies reporting statistical results (thus excluding case reports [49]), 31 used a one-group pretest-posttest design, while six employed a parallel or crossover design, of which four were randomized, and two were not. Regarding preclinical studies, 25 of the 27 (93%) used a parallel design, and no concurrent pharmacotherapy.

The heterogeneity of NIBS study protocols and participant characteristics were also assessed. Large differences across study protocols was the number of sessions, biomarker sampling time-points and, for TMS, stimuli schedule (see Appendix A). In clinical studies (excluding case reports [49]), 30 of 37 treated patients diagnosed with depression (Appendix A). The majority of clinical studies recruited more females than males (25 of the 35 provided the gender ratio, 71%) or recruited equal amounts (two, 6%), with the age mean or median ranging between 40 and 60 (with 27 of the 36 offering such information, 72%). By contrast, in all but two preclinical studies providing sex ratios, animals were all males. Most preclinical studies used healthy animals (22 of 27, 81%). Of 27 preclinical studies, 24 (89%) used rats, two (7%) mice, and one study used dogs (4%). Lastly, biomarker sampling time points appeared arbitrarily chosen or is vaguely described. For a more detailed discussion of these heterogenous methods of our included studies, see Appendix A.

### 3.3. Qualitative Synthesis Results

Figure 4a,b and Figure 5a–f visualize results of the counts of experimental groups with significant or nonsignificant changes in biomarkers of TRP metabolism, with the latter organized according to sampling medium. Figure 5a–c show examinations of CSF (a), plasma (b), serum (c) in clinical studies, and Figure 5d–f show brain tissue (d), microdialysis (e), and plasma (f) in preclinical studies. The sampling results of brain tissue and microdialysis involved various regions but are collapsed in Figure 5d and e for brevity; they are shown individually in Appendix A. Moreover, for brevity, only ECT and TMS results are visualized, and readers are referred to the bottom of Table 2 and Table 3 for the single clinical or preclinical tDCS study, respectively. For a more detailed report on the qualitative analysis of the methods and results in clinical and preclinical studies, see Appendix A.

### 3.4. Quantitative Synthesis Results by Bayesian Meta-Analysis

Table 4 lists the BF10 values of effects, heterogeneity, and publication bias for the changes in the biomarker levels by NIBS, where r is set to 0.5, 0.7, or 0.9, in addition to the counts of experimental groups used for calculation. Here, only the outcomes with at least moderate evidence for effects or null effects (BF10 > 3 or <1/3, respectively) across the three r’s are shown. The rest are available in the Appendix A. Consistent with our observations from the qualitative analysis, there was strong evidence of heterogeneity of studies assessing ECT effects on TRP plasma levels (BF10>10 across the three r’s), based on the eight studies with calculable SMDs [35,39,43,48,52,53,56,58]. However, these studies also gave moderate evidence for null effects. Furthermore, the same two studies as in the qualitative analysis provided moderate evidence for TMS increasing serum levels of 5-HT (Appendix A) [62,63].

Null effects on three other biomarkers were also observed. Allen et al. [35] and Ryan et al. [52] gave moderate evidence for no change in plasma KA levels and KA/KYN ratio in response to ECT. Both studies were multi-session and recruited depressed patients, with the timing of post-ECT sampling on the scale of days (4–7 days in Allen et al. [35] and 1–3 days in Ryan et al. [52]). Furthermore, Sibon et al. [67] and Tateishi et al. [68] provided moderate evidence for no effects on plasma levels of TRP in response to TMS. However, while Tateishi et al. [68] applied multiple treatment sessions on depressed patients, Sibon et al. [67] entailed one session with young and healthy volunteers. Neither study offered the timing of post-TMS sampling.

SMDs for changes in the levels of the biomarkers were calculable in 32 of 37 clinical studies (86%, excluding case reports [49]), while SMDs for changes in the clinical measures were calculable for 22 (59%; values available in the Appendix A). Compared to the effects of NIBS on biomarker concentrations, the BF10 values for ECT and TMS improving clinical measures were both larger than 100, indicating high probably of therapeutic outcomes in response to NIBS (other data are available in the Appendix A).

## 4. Discussion

We conducted a scoping review, which conformed with the PRISMA-ScR statement [27], with qualitative and meta-analysis to assess effects of therapeutic NIBS on biomarkers of TRP metabolism. We used the RoBMA [28] to synthesize the findings of studies with calculable SMDs (Table 4) and found moderate evidence for no effects on plasma TRP levels following ECT, with strong evidence for heterogeneity of these results. These findings suggest the absence of an effect by ECT on biomarkers of interest, but these results are inconclusive as they are more-than-not likely due to differences between ECT protocols or varied time points of post-ECT biomarker sampling. We found moderate evidence for TMS elevating effects on serum levels of 5-HT [62,63], but no effects on plasma KA levels and KA/KYN ratio post-ECT [35,52], and no effects on plasma TRP following TMS [67,68]—however, these latter findings were based on two studies each. Overall, the outcomes of NIBS on these biomarkers of interest (Figure 1) are highly heterogenous, with most studies finding null effects across biomarkers (Figure 4) and sampling method (Figure 5), and reporting insufficient statistics for quantitative synthesis, characterized by high methodological heterogeneity, with only a small number using adequately randomized controlled designs (Appendix A). In summary, we cannot conclusively claim or rule out the effects of NIBS on TRP metabolism given the current literature. However, taken at face-value, our findings provide preliminary evidence that informs current hypotheses for the mechanism of action by therapeutic NIBS across neuropsychiatric disorders, specifically mechanisms influenced by TRP metabolism and its molecular targets. To wit, below we discuss our findings in the context of evidence of NIBS’s impact on substrate availability and cytokine expression that mediate TRP metabolism, followed by changes to neurotransmitter potency critical to the induction of neuroplasticity.

As many studies on NIBS conclude, its mechanism of action need further investigation, but current evidence suggests that the observed therapeutic effects may be underlined by changes to long-term potentiation (LTP) and long-term depression (LTD) induction throughout the CNS. Evidence for these NIBS effects, in association with effects on TRP metabolism, are findings of stimulation-induced molecular and morphological changes in the CNS. For example, Peng et al. [94] investigated the effects of low to high frequency (HF) rTMS over the vertex of unpredictable-stress treated rats, which elevated prefrontal catecholamine levels and reduced MAO-A activity. HF-rTMS at 5 Hz had the highest elevating effects on prefrontal 5-HT levels, while also reducing 5-HIAA, and MAO-A activity [94]—this latter protein being critical to the degradation of 5-HT to 5-HIAA (Figure 1). Our meta-analysis showed elevated serum 5-HT levels following TMS, with null or mixed findings across NIBS and biomarker sampling methods. Indeed, clinical and preclinical studies that assessed the effects of TMS [62,63,64,65,68,69,83,84,85,86,87,88,89,90,91,92,93,94,95,96] and ECT [44,57,71,72,73,74,75,76,77,78,79,80] on 5-HT concentrations reported mostly heterogenous results, with 61% of comparisons reporting null effects (Figure 4a,b), with similar mixed results when examining effects by biomarker source (Figure 5a–f). ECT did not have a significant effect on 5-HT levels, suggesting that the therapeutic mechanisms of ECT differ from rTMS. Indeed, in a recent PET study examining TRD patients treated with ECT, Baldinger-Melich et al. [25] reported negligible changes to MAO-A expression in the cerebral cortex of patients after compared to before ECT, despite high antidepressant effects. Regarding the kynurenines, a recent pilot study examined the effects of rTMS on inflammatory cytokines, finding no effects; nor did cytokine levels correlate with depression severity, although significant antidepressants effects were observed [126]. Our results are consistent with these findings, as null effects of rTMS on KYN/TRP (Figure 4a), a biomarker for the rate of TRP metabolism toward the kynurenine pathway, were observed. Such findings suggest that the therapeutic effects of rTMS may not involve changes to inflammatory cytokine expression that lead to increased activation of the kynurenine pathway.

An alternative and exploratory link between NIBS and TRP metabolism is the research thread on the stimulation effects on the autonomic nervous system (ANS). Heart rate variability is a proxy of ANS health, with high variability interpreted as indicating abnormal ANS functioning in depressed patients [127,128]. Excitatory stimulation of the left DLPFC, a common therapeutic rTMS protocol, has restorative effects on this variability compared to sham stimulation [129]. As absorption in the small intestine is also a function of the ANS, the therapeutic effects of rTMS may affect TRP absorption. A consequence of this hypothesis is abnormal TRP levels in peripheral circulation following abnormal ANS function. Indeed, peripheral TRP levels are significantly lower in depressed patients compared to healthy controls [9]. Restoration of ANS function in depressed patients by NIBS would hypothetically increase peripheral TRP levels if symptoms and these biomarkers are associated—however, our included studies using rTMS mostly found null effects on TRP levels [59,60,69], with one finding significant elevated levels in plasma [68], and meta-analysis showing moderate evidence for no effects of ECT [35,39,43,48,52,53,56,58], despite the antidepressant efficacy of these techniques. This is an interesting line of research, and more direct experiments testing whether there is an association between ANS functioning and TRP availability are needed. The effects on TRP availability are critical, as this is the main source of substrate for the 5-HT and KYN pathways, critical for neuroplasticity [15], and for immune response mediation [1,2,7].

Despite the ubiquitous distribution and functional roles of 5-HT, most of its TRP substrate that enters the CNS is degraded toward the kynurenines (Figure 1). Various neuroactive metabolites are produced by this pathway, such as KA and QA, as they are not permeable across the blood–brain barrier—although, peripheral KYN may also be a major source for KA and QA in the CNS, as KYN has been observed to be highly permeable to the blood-brain barrier [130]. In the CNS, KA and QA have various roles, including in immune response, as both act on microglia. These have further effects on neuroplasticity, as KA acts as a receptor antagonist on neuronal N-methyl-D-aspartate (NMDA) whereas QA acts as an agonist, and with further mechanisms of each altering glutamate levels differentially. The effects of KA are typically considered neuroprotective, whereas QA effects are neurotoxic [7]. In support of these effects, low levels of the KA/QA ratio, indicating more neurotoxic and less neuroprotective products from the kynurenine pathway, were observed to coincide with more severe anhedonia symptoms and smaller cortical thickness of the right dorsal anterior cingulate cortex (ACC) in untreated depressed patients compared to controls [131]. In patients with bipolar disorder, KA/QA ratio has also been observed to be significantly lower than healthy controls [132]. More recently, a meta-analysis sought to identify which biomarkers of kynurenine pathway activity differentiate psychiatric disorders, finding KA/QA being significantly lower in MDD and bipolar disorder patients compared to healthy controls [9]. However, despite ECT having significant antidepressant effects (in uni- and bipolar patients) [17], two included studies reported null effects of ECT on QA/KA, an inversed form [33,52], but a third reported significant reductions of the QA/KA ratio [54]—suggesting less neurotoxic QA and more neuroprotective KA, and no studies investigating these ratios after tDCS nor rTMS (Figure 4). Consistent with these effects, our meta-analysis indicated no effects on plasma KA levels and KA/KYN ratio following ECT [35,52]. Moreover, two studies using rTMS found null effects on KA levels in plasma and CSF despite antidepressant effects [68,69]. This reinforces the possibility that the antidepressant effects of NIBS may not be associated with changes to the production of these neuroactive kynurenines but may instead target downstream molecular targets of TRP metabolism.

NIBS has been observed to potentiate signaling strength of TRP metabolites. For instance, in a PET study, HF-rTMS had significant antidepressant effects, with changes to symptom severity positively correlated with change in 5-HT_2A_ receptor binding in bilateral DLPFC, but negatively correlated with right hippocampus receptor binding [133]. This receptor is thought to mediate expression of brain derived neurotrophic factor (BDNF), with 5-HT_2A_ agonists decreasing BDNF mRNA expression in the hippocampus, but increasing expression in the rat neocortex [134]. In another PET study on healthy participants, the effects of HF-rTMS over the left DLPFC on limbic serotonin synthesis were assessed with the radioligand [(11)C]-alpha-methyl-tryptophan, thought to approximate the capacity of TRP metabolism toward the 5-HT pathway. Brain stimulation over the left DLPFC, compared to over the left occipital cortex, was followed by increases in 5-HT release in the right posterior cingulate cortex, but not cortical or sub-cortical regions, with the metabolite presumed to be from terminals of dorsal raphe nuclei [67]. Baldinger et al. [135] reviewed the literature on the effects of ECT on 5-HT neurotransmission, reporting mixed effects on 5-HT_1A_, including no effects [136] and diminished 5-HT_1A_ receptor binding in the right subgenual ACC, orbitofrontal cortex, amygdala, and hippocampus [137]. In a separate PET study, 5-HT_2A_ receptor binding was also found to be globally diminished, including in the bilateral occipital cortex, the medial parietal cortex, the limbic cortex, and the bilateral prefrontal cortex of depressed patients after ECT [138]. Thus, while the effects of NIBS on 5-HT and other biomarkers of TRP metabolism may not be directly observable, as indicated by our findings, modulations of the molecular targets of these metabolites may instead underlie the therapeutic effects of NIBS. 

We observed several limitations with our review methods. First, biomarker keywords were not abbreviated in our search, nor were all names for the same metabolite used. Potentially, this may have caused some studies to have been missed. We attempted to mitigate this by screening the references of reviews and included studies. Second, some metabolites that may be relevant were excluded (such as 6-sulfatoxymelatonin, a product of melatonin metabolism—e.g., [105] was excluded; Appendix A). We initially aimed to keep this review focused on TRP metabolites commonly discussed in light of their vital physiological roles, such as 5-HT, melatonin, kynurenine, and NAD+ [7]. Likewise for NIBS, not all techniques were considered, such as transcranial electric stimulation, though several more exist beyond those reviewed here [22]. We sought studies using ECT, rTMS, and tDCS as these are frequently discussed in reviews on the therapeutic efficacy of NIBS and mechanisms of action, e.g., [17,139]. Lastly, our univariate focus on biomarker levels may not be sufficient to elucidate the mechanisms of action by NIBS and the relation between TRP metabolism and health condition—this aim will require future studies to utilize cross-domain expertise, including biomarker effects. That is, research on the dynamics within and between multiple levels of analysis are needed to develop etiologies of neuropsychiatric disorders that capture real-world variability of patients [140] and to elucidate the dynamics between adaptive biological architecture and metabolic homeostasis. For instance, whether change in brain activity and noise following successful therapeutic NIBS coincides to changes in biomarker levels is an important research question to be investigated with careful signal analysis approaches, such as [141,142,143].

## 5. Conclusions

This scoping review investigated the effects of non-invasive brain stimulation on biomarkers of tryptophan metabolism and synthesized relevant studies using qualitative and meta-analyses. Although the amount of evidence for each biomarker in clinical and preclinical studies is sparse, we were able to conduct meta-analyses, although studies eligible for this synthesis were few. In agreement with Bayesian meta-analysis results, qualitative analysis revealed highly heterogenous methods and findings in this literature. Thus, more randomized controlled studies are needed to elucidate the effects of therapeutic non-invasive brain stimulation on tryptophan metabolism and the correspondence between biomarkers and abnormalities observed in several neuropsychiatric disorders. Such findings would inform optimal use of brain stimulation, in concurrence with other therapies. Specifically, as the nervous system functions by electrochemical principles, the optimal application or synergism of electrical (e.g., non-invasive and invasive brain stimulation) and chemical (e.g., antidepressant and anti-inflammatory treatments) therapies need further investigation.

## Figures and Tables

**Figure 1 ijms-23-09692-f001:**
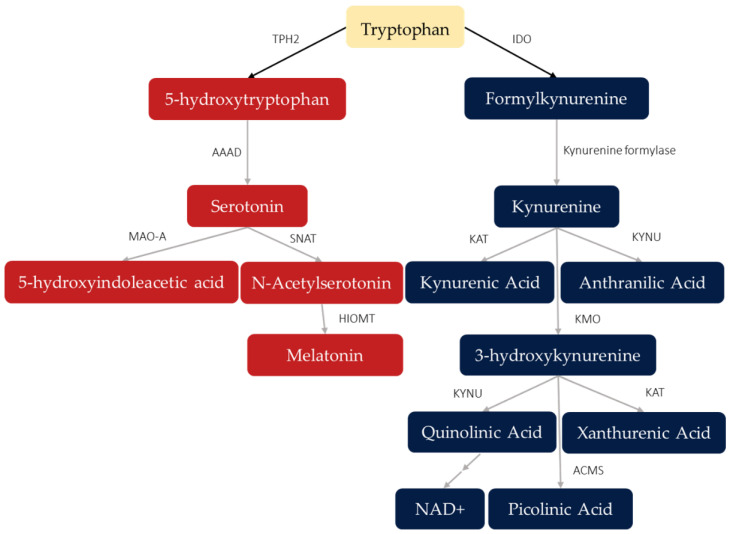
Products of tryptophan metabolism via serotonin (red) and kynurenine (blue) pathways. Arrows point toward the direction of metabolism as mediated by different enzymes or other catalysts, with fields showing metabolites produced along these pathways. Double arrows indicate multiple metabolic steps (catalysts not shown). Abbreviations: TPH2: tryptophan hydroxylase, isoenzyme 2; AAAD: aromatic acid decarboxylase; MAO-A: monoamine oxidase A; HIOMT: hydroxyindole O-methyl transferase; SNAT: serotonin-N-acetyltransferase; IDO: indoleamine 2,3-dioxyenase; KAT: kynurenine aminotransferase; KMO: kynurenine 3-monooxygenase; KYNU: kynurinase; ACMS: 2-amino-3-carboxymuconic-6-semialdehyde decarboxylase.

**Figure 2 ijms-23-09692-f002:**
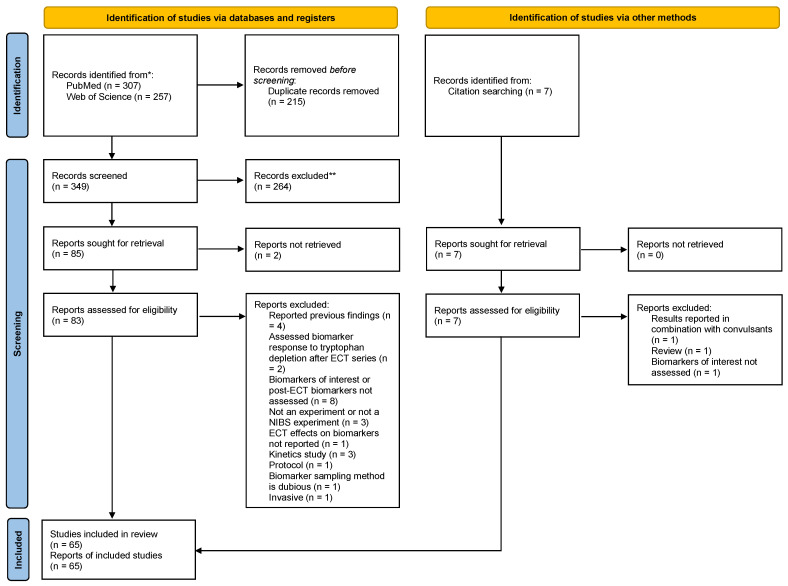
PRISMA flowchart. *: for each database in our search; **: no automation tools were used.

**Figure 3 ijms-23-09692-f003:**
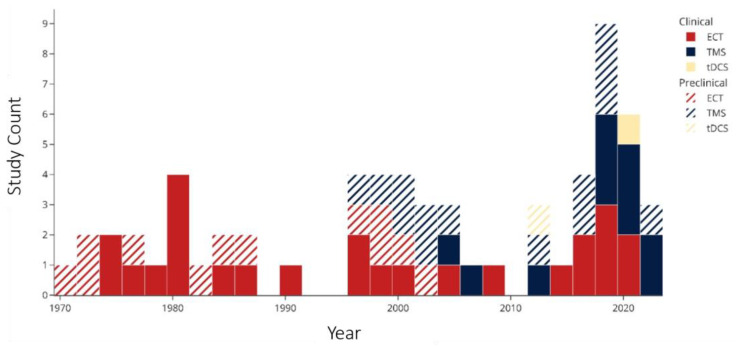
Publication year of included studies. Abbreviations: ECT: electroconvulsive therapy; TMS: transcranial magnetic stimulation; tDCS: transcranial direct current stimulation.

**Figure 4 ijms-23-09692-f004:**
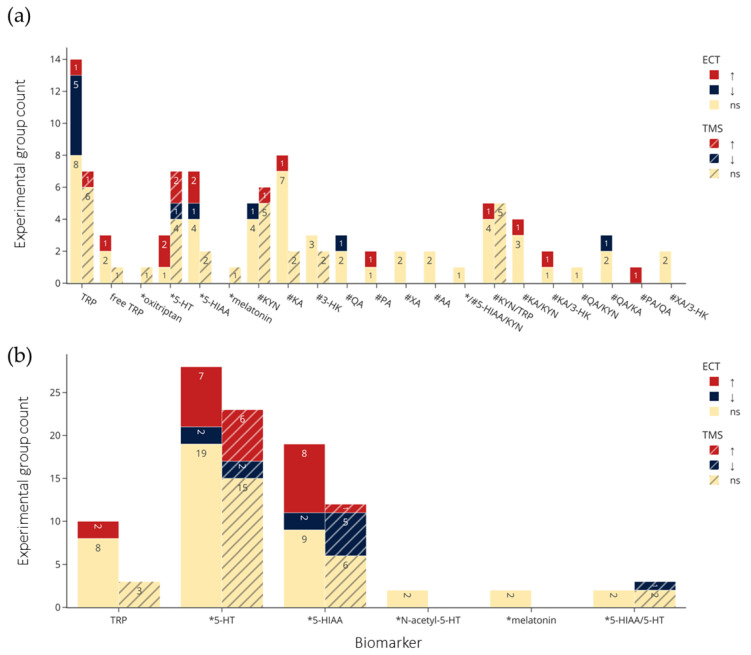
Summary of (**a**) clinical and (**b**) preclinical trials findings, including direction of effects by NIBS and by biomarker. Colors indicate statistical effects on biomarkers by NIBS: red = significant increase; blue = significant decrease; yellow = no significant changes to biomarker levels following NIBS. ↑: Significantly increased or the experimental group levels were significantly larger than the control group; ↓: significantly decreased or the experimental group levels significantly smaller than the control group, *: biomarkers in the serotonin pathway; #: biomarkers in the kynurenine pathway. TRP refers to total TRP when not specified, and for brain tissue and microdialysis the regions are collapsed. Abbreviations: TRP: tryptophan; 5-HT: serotonin; 5-HIAA: 5-hydroxyindoleacetic acid; KYN: kynurenine; KA: kynurenic acid; 3-HK: 3-hydroxykynurenine; QA: quinolinic acid; XA: xanthurenic acid; AA: anthranilic acid; PA: picolinic acid; ECT: electroconvulsive therapy; TMS: transcranial magnetic stimulation; ns: not significant.

**Figure 5 ijms-23-09692-f005:**
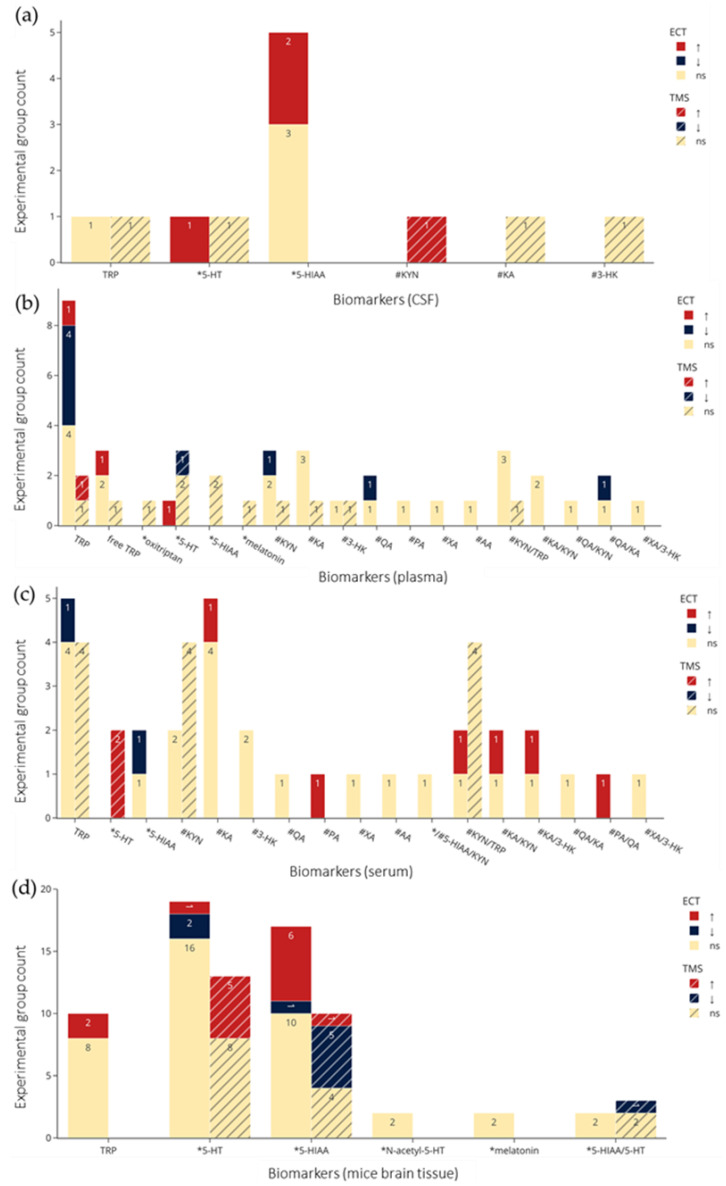
Count of experimental groups reporting significant increases, decreases, or non-significant changes in the metabolites by NIBS. (**a**–**c**) From CSF, plasma, and serum in clinical studies respectively. (**d**–**f**) from brain tissue, microdialysis, and plasma in preclinical studies respectively. Colors indicate statistical effects on biomarkers by NIBS: red = significant increase; blue = significant decrease; yellow = no significant changes to biomarker levels following NIBS. *: biomarkers in the serotonin pathway; #: biomarkers in the kynurenine pathway. TRP refers to total TRP when not specified, and for brain tissue and microdialysis the regions are collapsed. Abbreviations: TRP: tryptophan; 5-HT: serotonin; 5-HIAA: 5-hydroxyindoleacetic acid; KYN: kynurenine; KA: kynurenic acid; 3-HK: 3-hydroxykynurenine; QA: quinolinic acid; XA: xanthurenic acid; AA: anthranilic acid; PA: picolinic acid; ECT: electroconvulsive therapy; TMS: transcranial magnetic stimulation; ns: not significant.

**Table 1 ijms-23-09692-t001:** PICO statement.

**P**atient	Human or animal models; can be healthy or with an underlying health condition.
**I**ntervention	NIBS techniques: ECT, rTMS, or tDCS
**C**omparison	Biomarkers of interest assessed: tryptophan (TRP), kynurenine (KYN), formylkynurenine, kynurenic acid (KA), quinolinic acid (QA), NAD+, 3-hydroxykynurenine (3-HK), xanthurenic acid (XA), picolinic acid (PA), anthranilic acid (AA), serotonin (5-HT), oxitriptan, 5-hydroxytryptamine (5-HT), 5-hydroxyindoleacetic acid (5-HIAA), N-acetyl-5-HT, or melatonin. Measures of their concentrations must be reported and compared to baseline levels, e.g., measures obtained before and again during or after NIBS, or comparison of animal models that received active versus sham NIBS. Studies without a control group can be included, so long as baseline measures are clearly defined.
**O**utcome	Biomarkers must measure serotonin or kynurenine pathway activation, sampled peripherally or centrally (e.g., CSF, plasma/serum, microdialysis, brain tissue analysis in animal models). Changes in health condition will be collected if available (e.g., in humans, clinical scales; in rodents, task specific performance)

**Table 2 ijms-23-09692-t002:** Clinical studies included: characteristics and individual results.

Study	Design	Protocol	Participants	Demographics	Source	Timing of Post-NIBS Sampling	Biomarker	Result
ECT								
Aarsland et al., 2019 [33]	pretest-posttest	**Site**: right unilateral **Seizure length**: *median 50.7 s, IQR 16 s **Session N**: *max. 20 **Session freq**.: 3/week	depression (uni- and bipolar, with and without psychotic symptoms)	**N**: 21 **M:F**: 44:56% * **Age**: median 46, IQR 21.0 *	serum	1–2 weeks after the last session (median 10 days, IQR 6)	TRP	ns
							KYN	ns
							KA	ns
							3-HK	ns
							QA	ns
							PA	↑
							XA	ns
							AA	ns
							KYN/TRP	ns
							KA/KYN	ns
							KA/3-HK	ns
							QA/KA	ns
							XA/3-HK	ns
							PA/QA	↑
Aberg-Wistedt et al., 1986 [34]	pretest-posttest	**Site**: bifrontotemporal **Seizure length**: 45.0 ± 8.2 s * **Session N**: 8.0 ± 3.1 * **Session freq**.: 3/week	depression (unipolar, without psychotic symptoms)	**N**: 11 **M:F**: 33:67% * **Age**: 56 ± 16.2 *	CSF	3 weeks after the last session	5-HIAA	ns
Allen et al., 2018 [35]	pretest-posttest	**Site**: bitemporal **Session N**: max. 6 **Session freq**.: 2/week	TRD (unipolar)	**N**: 18 **M:F**: 33:67% **Age**: 57.5 ± 14.7	plasma	4–7 days after the last session	TRP	ns
							KYN	ns
							KA	ns
							KYN/TRP	ns
							KA/KYN	ns
D’Elia et al., 1977 [36]	pretest-posttest	**Site**: nondominant unilateral **Session N**: mean 6.1, range 3–12	depression (uni- and bipolar)	**N**: 24 **M:F**: 38:63% **Age**: 48.0 ± 12.5, range 22–64	serum	mean of levels sampled before each session from the second to the last.	TRP	ns
Guloksuz et al., 2015 [37]	pretest-posttest	**Site**: bifrontotemporal **Seizure length**: mean 52 s, range 10–200 s **Session N**: mean 6.1, range 3–11 **Session freq**.: 2/week	TRD (uni- and bipolar)	**N**: 19 **M:F**: 32:68% **Age**: 52.6 ± 14.4, range 23–74	serum	before every other session	TRP	ns
							KYN	ns
							KA	↑
							3-HK	ns
							5-HIAA	ns
							KYN/TRP	↑
							KA/KYN	↑
							KA/3-HK	↑
							5-HIAA/KYN	ns
Hasani et al., 2019 [38]	parallel	**Site**: bifrontotemporal **Session N**: 6–8 **Session freq**.: every other day	TRD (unipolar)	**Exp. group N**: 21 **Age**: 35.24 ± 14.17 **Control group N**: 15 **Age**: 33.26 ± 13.59	plasma	16 days after the first session	5-HT	↑ (posttest)
						30 days after the first session		↑ (posttest)
Hoekstra et al., 2001 [39]	pretest-posttest	**Site**: unilateral and bifrontotemporal **Session N**: mean 11 **Session freq**.: 2/week	TRD (unipolar, with and without psychotic symptoms)	**N**: 20 **M:F**: 35:65% **Age**: 52 ± 13.1	plasma	the morning after the final session	TRP	ns
Hoffmann et al., 1985 [40]	pretest-posttest	**Site**: bitemporal **Session N**: mean 11, range 6–19 * **Session freq**.: 3/week	depression (uni- and bipolar, with and without psychotic symptoms)	**N**: 7 all male **Age**: 46 ± 11.2, range 29–61	CSF	12 days after the last session	5-HIAA	ns
Hofmann et al., 1996 [41]	pretest-posttest	**Site**: bifrontotemporal **Seizure length**: 40.48 ± 14.12 s **Session N**: 1	depression (unipolar)	**N**: 31 **M:F**: 29:71% **Age**: 37.22 ± 13.98	serum	1 min after the session	5-HIAA	ns
						1 h after the session		↓
						2 h after the session		↓
Jori et al., 1975 [42]	pretest-posttest	**Session N**: 4–8	depression (with psychotic symptoms)	**N**: 12 **M:F**: 88:12% * **Age**: mean 45, range 25–63 *	CSF	7 days after the final session	5-HIAA	↑
Kirkegaard, Møller and Bjørum, 1978 [43]	pretest-posttest	**Site**: unilateral **Session N**: 10.5 ± 2.6 **Session freq**.: 2/week	depression (uni- and bipolar)	**N**: 10 **M:F**: 30:70% **Age**: mean 63	CSF	-	TRP	ns
					plasma		TRP	↓
							free TRP	ns
							free/total TRP	↑
Lestra et al., 1998 [44]	pretest-posttest	**Session N**: 5–9 **Session freq**.: every 3 or 4 days	depression (unipolar)	**N**: 6 **M:F**: 43:57% * **Age**: range 27–70 *	platelet pellet	10 days after the last session	5-HT	ns
Mokhtar et al., 1997 [45]	pretest-posttest	**Site**: bilateral **Seizure length**: 20 s in nine, a mild fit in one **Session N**: 1	depression (uni- and bipolar)	**N**: 10 **M:F**: 50:50% **Age**: mean 48.5, SEM 4.3	serum	15 min after the session	TRP	ns
						30 min after the session		ns
						45 min after the session		↓
						1 h after the session		↓
Nikisch and Mathé, 2008 [46]	pretest-posttest	**Site**: unilateral **Session N**: 8 **Session freq**.: 2/week	TRD (unipolar, with and without psychotic symptoms)	**N**: 6 **M:F**: 83:17% **Age**: 40 ± 8	CSF	1 week after the last session	5-HIAA	↑
Olajossy et al., 2017 [47]	pretest-posttest	**Site**: bifrontotemporal **Session N**: 12 **Session freq**.: 2/week	recurrent depressive disorder	**N**: 32 **M:F**: 41:59% **Age**: 49.41 ± 12.73	serum	30 min after the first, sixth, and 12th sessions	KA	ns
			depression in bipolar disorder	**N**: 11 **M:F**: 36:64% **Age**: 44.73 ± 13.83				ns
			schizoaffective disorder	**N**: 7 **M:F**: 43:57% **Age**: 33.29 ± 8.56				ns
Palmio et al., 2005 [48]	pretest-posttest	**Site**: bilateral **Seizure length**: 45.7 ± 13.2, range 21–66 **Session N**: 3.8 ± 2.2, range 1–7	depression (unipolar, with and without psychotic symptoms)	**N**: 10 **M:F**: 30:70% **Age**: 55.6 ± 15.8, range 28–73	plasma	2 h after the last session	TRP	↑
						6 h after the last session		↑
						24 h after the last session		↑
						48 h after the last session		ns
Rudorfer et al., 1988 [49]	pretest-posttest	**Site**: uni- and bilateral **Seizure length**: at least 25 s **Session N**: 14.3 ± 1.2, range 13–15	depression (unipolar)	**N**: 3 all female **Age**: 48.7 ± 22.0, range 35–74	CSF	4–5 days after the last session	5-HIAA	No statistical tests: levels reportedly increased in 2 of 3 patients.
Rudorfer et al., 1991 [50]	pretest-posttest	**Site**: uni- and bilateral **Seizure length**: at least 25 s **Session N**: 13.6 ± 3.3, range 9–19	TRD (unipolar)	**N**: 8 **M:F**: 25:75% **Age**: 53.1 ± 13.2, range 35–71	CSF	5–10 days after the last session	5-HIAA	ns
Ryan et al., 2020 [51]	pretest-posttest	**Site**: unilateral and bitemporal **Session N**: 7.97 ± 2.46 **Session freq**.: 2/week	depression (uni- and bipolar, with and without psychotic symptoms)	**N**: 94 **M:F**: 38:62% **Age**: 55.48 ± 14.72	plasma	1–3 days after the last session	XA/3-HK	ns
Ryan et al., 2020 [52]	pretest-posttest	**Site**: unilateral and bitemporal **Session N**: 7.96 ± 2.47 **Session freq**.: 2/week	depression (uni- and bipolar, with and without psychotic symptoms)	**N**: 94 **M:F**: 38:62% **Age**: 55.48 ± 14.72	plasma	1–3 days after the last session	TRP	ns
							KYN	ns
							KA	ns
							3-HK	ns
							QA	ns
							PA	ns
							XA	ns
							AA	ns
							KYN/TRP	ns
							KA/KYN	ns
							QA/KA	ns
							QA/KYN	ns
Sawa, 1981 [53]	pretest-posttest	**Session N**: 1	depression (uni- and bipolar)	**N**: 9 **M:F**: 44:56% **Age**: 40.7 ± 11.8, range 20–52	plasma	1 min after the session	TRP	ns
							free TRP	↑
						5 min after the session	TRP	↓
							free TRP	ns
						10 min after the session	TRP	↓
							free TRP	ns
						30 min after the session	TRP	ns
							free TRP	ns
						1 h after the session	TRP	ns
							free TRP	↓
Schwieler et al., 2016 [54]	pretest-posttest	**Site**: right unilateral **Session N**: 3	TRD (unipolar)	**N**: 15 **M:F**: 58:42% * **Age**: median 41.0, IQR 25.0–54.0 *	plasma	-	TRP	↓
							KYN	↓ $
							KA	ns
							QA	↓
							KYN/TRP	ns
							QA/KA	↓
Smith and Strömgren, 1981 [55]	pretest-posttest	**Site**: nondominant unilateral **Session freq**.: 4/week	depression	**N**: 13 **M:F**: 69:31% **Age**: 42.7 ± 12.5, range 24–61	serum	9 am on the day after the last session	TRP	ns
Stelmasiak and Curzon, 1974 [56]	pretest-posttest	**Site**: unilateral **Session N**: 1	depression	**N**: 18 **M:F**: 39:61% **Age**: mean 48, range 20–70	plasma	1 min after the session	TRP	ns
							free TRP	↑
						15 min after the session	TRP	ns
							free TRP	↑
						30 min after the session	TRP	ns
							free TRP	ns
						1 h after the session	TRP	ns
							free TRP	ns
Udayakumar et al., 1981 [57]	pretest-posttest	**Site**: bitemporal **Session N**: 1	schizophrenia	**N**: 29 **M:F**: 66:34% **Age**: 27.4 ± 8.0, range 14–49	CSF	5 min after the session	5-HT	↑
						48–72 h after the session		ns
Whalley, Yates and Christie, 1980 [58]	pretest-posttest	**Session N**: 4–9 *	depression (unipolar, without psychotic symptoms)	**N**: 11 **M:F**: 25:75% * **Age**: 49 ± 13.9 *	plasma	10 min after the first session	TRP	↓
							free TRP	ns
						immediately before the last session	TRP	ns #
							free TRP	ns #
						12 weeks after the last session	TRP	ns $
							free TRP	ns $
**TMS**								
Leblhuber et al., 2021 [59]	pretest-posttest	**Site**: medial frontopolar cortex **Pulse freq**.: 20 Hz **Intensity**: 1.5 T **Pulse N**: 2400 **Session N**: 10 **Session freq**.: 5/week	TRD (unipolar)	**N**: 21 **M:F**: 48:52% **Age**: 59.4 ± 15.7	serum	-	TRP	ns
							KYN	ns
							KYN/TRP	ns
Leblhuber, Steiner and Fuchs, 2019 [60]	parallel	**Site**: bilateral prefrontal cortex **Pulse freq**.: 3 Hz **Intensity**: 0.08 T **Pulse N**: 30 min **Session N**: 10 **Session freq**.: 5/week	TRD (geriatric)	**Exp. group N**: 19 **M:F**: 45:55% * **Age**: mean 71.9, SEM 2.92 **Control group N**: 10 **M:F**: 45:55% * **Age**: mean 73.3, SEM 2.69	serum	immediately after the last session	TRP	ns (group x time)
							KYN	ns (group x time)
							KYN/TRP	ns (group x time)
Leblhuber et al., 2018 [61]	pretest-posttest	**Site**: bilateral prefrontal cortex **Pulse freq**.: 3 Hz **Intensity**: above 100% MT **Session N**: 10	TRD (geriatric)	**N**: 10 **M:F**: 40:60% **Age**: 69 ± 8.78	serum	-	TRP	ns
							KYN	ns
							KYN/TRP	ns
Liu et al., 2022 [62]	parallel	**Site**: bilateral dorsolateral and ventrolateral prefrontal cortex **Pulse freq**.: 0.5 Hz **Intensity**: 0.70 T **Session N**: 8 **Session freq**.: 5/week	post-stroke depression	**Exp. group N**: 35 **M:F**: 57:43% **Age**: 55.61 ± 6.84 **Control group N**: 35 **M:F**: 66:34% **Age**: 50.20 ± 6.28	serum	8 weeks after the last session	5-HT	↑ (posttest)
Lu et al., 2018 [63]	pretest-posttest	**Site**: bilateral dorsolateral prefrontal cortex **Pulse freq**.: 1 Hz **Intensity**: 80% MT **Pulse N**: 750 **Session N**: 10 **Session freq**.: 5/week	generalized anxiety disorders	**N**: 28 **M:F**: 39:61% **Age**: 45.5 ± 12.67, range 27–72	serum	1 h after the last session	5-HT	↑
Maestú et al., 2013 [64]	parallel	**Site**: 33 stimulation coils distributed evenly across an EEG cap **Pulse freq**.: 8 Hz **Intensity**: 43 nT **Pulse N**: 20 min **Session N**: 8 **Session freq**.: 1/week	fibromyalgia	**Exp. group N**: 28 all female **Age**: 40.7 ± 6.7 * **Control group N**: 26 all female **Age**: 40.7 ± 6.7 *	blood	-	5-HT	ns (group x time)
Miniussi et al., 2005 [65]	pretest-posttest	**Site**: left dorsolateral prefrontal cortex **Pulse freq**.: 17 Hz **Intensity**: 110% MT **Pulse N**: 408 **Session N**: 5 **Session freq**.: 1/day	TRD (uni- and bipolar, with and without psychotic symptoms)	**N**: 10 **M:F**: 25:75% * **Age**: mean 58	plasma	the day of the last session	5-HT	ns
							5-HIAA	ns
		**Pulse freq**.: 1 Hz **Pulse N**: 400		**N**: 10 **M:F**: 25:75% * **Age**: mean 52			5-HT	ns
							5-HIAA	ns
Niimi et al., 2020 [66]	parallel	**Site**: primary motor cortex of the unaffected hemisphere **Pulse freq**.: 1 Hz **Intensity**: 90% MT **Pulse N**: 1200 **Session N**: 22 **Session freq**.: 2/day	stroke	**Exp. group N**: 62 **M:F**: 66:34% **Age**: 62.3 ± 11.0 **Control group N**: 33 **M:F**: 52:48% **Age**: 66.2 ± 10.8	serum	-	TRP	ns
							KYN	ns
							KYN/TRP	ns
Sibon et al., 2007 [67]	crossover	**Site**: left dorsolateral prefrontal cortex **Pulse freq**.: 10 Hz **Intensity**: 90% MT **Pulse N**: 450 **Session N**: 1	healthy	**N**: 10 **M:F**: 50:50% **Age**: 24.7 ± 5.14, range 18–40	plasma	-	TRP	ns
							free TRP	ns
Tateishi et al., 2021 [68]	pretest-posttest	**Site**: left dorsolateral prefrontal cortex **Pulse freq**.: 10 Hz **Intensity**: 100% MT **Pulse N**: 1600 **Session N**: 30 **Session freq**.: 5/week	TRD (unipolar)	**N**: 13 **M:F**: 31:69% **Age**: 54.9 ± 14.3	plasma	-	TRP	↑
							oxitriptan	ns
							5-HT	↓
							melatonin	ns
							KYN	ns
							KA	ns
							3-HK	ns
							KYN/TRP	ns
Tateishi et al., 2022 [69]	pretest-posttest	**Site**: left dorsolateral prefrontal cortex **Pulse freq**.: 10 Hz **Intensity**: 100% MT **Pulse N**: 1600 **Session N**: 30 **Session freq**.: 5/week	TRD (unipolar)	**N**: 5 **Age**: 48.0 ± 13.9, range 30–72	CSF	-	TRP	ns
							5-HT	ns
							KYN	↑
							KA	ns
							3-HK	ns
**tDCS**								
Hadoush et al., 2021 [70]	pretest-posttest	**Site**: bilateral primary motor cortex and bilateral dorsolateral prefrontal cortex **Polarity**: anodal **Electrode size**: 25 cm^2^ **Intensity**: 1 mA **Duration**: 20 min **Session N**: 10 **Session freq**.: 5/week	Parkinson’s disease	**N**: 25 **M:F**: 76:24% **Age**: mean 61.5, range 30–80	serum	-	melatonin	↓

↑: significantly increased or the experimental group levels were significantly larger than the control group; ↓: significantly decreased or the experimental group levels were significantly smaller than the control group; ns: non-significantly changed or different; *, #, and $: data not available for the featured group but taken from a larger group it belonged to or a subgroup of it, e.g., who were followed up; -: not applicable or data not available; N: count; freq.: frequency; exp.: experimental. For studies using a parallel design, “(group x time)” denotes the result based on group x time interaction, and “(posttest)” group difference in the posttest. Here TRP refers to total TRP when unspecified.

**Table 3 ijms-23-09692-t003:** Preclinical studies included: characteristics and individual results.

Study	Design	Protocol	Subjects	Demographics	Source	Timing of Post-NIBS Sampling	Biomarker	Result
ECT								
Evans et al., 1976 [71]	parallel	**Site:** ear clips **Session N:** 1	healthy Sprague-Dawley rats	**Exp. group N:** - all male **Control group N:** - all male	brain tissue	30 min after the session	TRP	ns (posttest)
							5-HT	ns (posttest)
							5-HIAA	ns (posttest)
				**Exp. group N:** 5 **Control group N:** 5		1 h after the session	5-HT	ns (posttest)
				**Exp. group N:** 6 **Control group N:** 5		3 h after the session	TRP	ns (posttest)
				**Exp. group N:** 14 **Control group N:** 14			5-HT	ns (posttest) *
							5-HIAA	↑ (posttest)
				**Exp. group N:** 3 **Control group N:** 3		6 h after the session	5-HT	ns (posttest)
				**Exp. group N:** 4 **Control group N:** 5		24 h after the session	TRP	ns (posttest)
							5-HT	ns (posttest)
							5-HIAA	ns (posttest)
Gur et al., 2002 [72]	parallel	**Site:** ear clips **Session N:** 10 **Session freq.:** 1/day	healthy Albino rats	**Exp. group N:** 13 all male **Control group N:** 16 all male	microdialysis (ventral hippocampus)	48 h after the last session	5-HT	ns (posttest)
					microdialysis (anterior hypothalamus)	24 h after the last session		ns (posttest)
Juckel et al., 1999 [73]	pretest-posttest	**Site:** left medial prefrontal cortex **Session N:** 1	healthy, anesthetized Sprague-Dawley rats	**N:** 6 all male	microdialysis (left amygdala)	during the session	5-HT	↑
		**Site:** left medial prefrontal cortex	healthy, anesthetized Sprague-Dawley rats	**N:** 6	microdialysis (left ventral hippocampus)			↑
		**Site:** left medial prefrontal cortex	healthy, behaving Sprague-Dawley rats	**N:** 3	microdialysis (left ventral hippocampus)			↑
		**Site:** right medial prefrontal cortex	healthy, anesthetized Sprague-Dawley rats	**N:** 4	microdialysis (left ventral hippocampus)			↑
		**Site:** right medial prefrontal cortex	healthy, anesthetized Sprague-Dawley rats	**N:** 3	microdialysis (right ventral hippocampus)			↑
Karoum et al., 1986 [74]	pretest-posttest	**Site:** ear clips **Session N:** 10 **Session freq.:** 1/day	healthy Sprague-Dawley rats	**N:** 5 all male	brain tissue (caudate nucleus)	20 h after the last session	5-HT	ns
							5-HIAA	ns
							5-HIAA/5-HT	ns
						1 week after the last session	5-HT	ns
							5-HIAA	ns
							5-HIAA/5-HT	ns
					brain tissue (frontal cortex)	20 h after the last session	5-HT	ns
							5-HIAA	ns
							5-HIAA/5-HT	ns
						1 week after the last session	5-HT	↑
							5-HIAA	ns
							5-HIAA/5-HT	ns
					urine	nights between sessions	5-HIAA	↓
						nights in the first week after the last session		↓
Khanna et al.,, 1971 [75]	parallel	**Site:** temples **Session N:** 15 **Session freq.:** 1/day	healthy dogs later with ECT-induced cardiac abnormalities	**Exp. group N:** 19 **Control group N:** 10	myocardium tissue	the day after the last session	5-HT	↑ (posttest)
			healthy dogs later without ECT-induced cardiac abnormalities	**Exp. group N:** 11 **Control group N:** 10				ns (posttest)
Madhav et al., 2000 [76]	parallel	**Site:** ear clips **Session N:** 5 **Session freq.:** every other day	Sprague-Dawley rats with serotonergic lesion in the right cingulum bundle (via which the hippocampus receives serotonergic innervation)	**Exp. group N:** 4 all male **Control group N:** 4 all male	brain tissue (right hippocampus)	18 days after the last session	5-HT	ns (posttest)
							5-HIAA	↑ (posttest)
							5-HIAA/5-HT	ns (posttest)
McIntyre and Oxenkrug, 1984 [77]	parallel	**Site:** ear clips **Seizure length:** 20–25 s **Session N:** 7 **Session freq.:** 1/day	healthy Sprague-Dawley rats	**Exp. group N:** 14 all male **Control group N:** 7 all male	brain tissue (hypothalamus)	9 pm of the day of the last session	TRP	ns (posttest)
							5-HT	↓ (posttest)
							5-HIAA	ns (posttest)
				**Exp. group N:** 13 **Control group N:** 7	brain tissue (pineal gland)		TRP	ns (posttest)
							5-HT	↓ (posttest)
							5-HIAA	ns (posttest)
							N-acetyl-5-HT	ns (posttest)
							melatonin	ns (posttest)
		**Session N:** 1		**Exp. group N:** 8 **Control group N:** 7	brain tissue (hypothalamus)		TRP	ns (posttest)
							5-HT	ns (posttest)
							5-HIAA	↓ (posttest)
				**Exp. group N:** 7 **Control group N:** 7	brain tissue (pineal gland)		TRP	ns (posttest)
							5-HT	ns (posttest)
							5-HIAA	ns (posttest)
							N-acetyl-5-HT	ns (posttest)
							melatonin	ns (posttest)
Shields, 1972 [78]	parallel	**Session N:** 8	healthy rats	**Exp. group N:** 8 **Control group N:** 8	brain tissue	24 h after the last session	5-HT	ns (posttest)
							5-HIAA	ns (posttest)
				**Exp. group N:** 8 **Control group N:** 7			5-HT	ns (posttest)
							5-HIAA	ns (posttest)
		**Session N:** 12		**Exp. group N:** 7 **Control group N:** 8			5-HT	ns (posttest)
							5-HIAA	ns (posttest)
		**Session N:** 16		**Exp. group N:** 6 **Control group N:** 8			5-HT	ns (posttest)
							5-HIAA	ns (posttest)
		**Session N:** 1		**Exp. group N:** 8 **Control group N:** 8		3 h after the session	TRP	ns (posttest)
							5-HT	ns (posttest)
							5-HIAA	↑ (posttest) *
Sugrue, 1983 [79]	parallel	**Site:** ear clips **Session N:** 10 **Session freq.:** 1/day	healthy Sprague-Dawley rats	**Exp. group N:** at least 5 all male **Control group N:** 40–48 all male	brain tissue (cortex)	6.5 h after the last session	5-HT	ns (posttest)
							5-HIAA	↑ (posttest)
Tagliamonte et al., 1972 [80]	parallel	**Site:** ear clips **Seizure length:** 15 s **Session N:** 2 **Session freq.:** 30 min interval	healthy Wistar rats	**Exp. group N:** 24 all male **Control group N:** 150 all male	plasma	1 h after the first session	TRP	ns (posttest)
					brain tissue		TRP	↑ (posttest)
							5-HT	ns (posttest)
							5-HIAA	↑ (posttest)
		**Session N:** 1		**Exp. group N:** 21 **Control group N:** 150	plasma		TRP	ns (posttest)
					brain tissue		TRP	↑ (posttest)
							5-HT	ns (posttest)
							5-HIAA	↑ (posttest)
Yoshida et al., 1997 [81]	parallel	**Site:** ear clips **Seizure length:** 20–25 s **Session N:** 8 **Session freq.:** 1/day	healthy Wistar rats	**Exp. group N:** 6 all male **Control group N:** 6 all male	microdialysis (striatum)	0–180 after the first session	5-HIAA	↑ (posttest/pretest)
						immediately before the last session		↑ (posttest)
Yoshida et al., 1998 [82]	parallel	**Site:** ear clips **Seizure length:** 20–25 s **Session N:** 8 **Session freq.:** 1/day	healthy Wistar rats	**Exp. group N:** 6 all male **Control group N:** 6 all male	microdialysis (frontal cortex)	0–180 after the first session	5-HIAA	↑ (posttest/pretest)
						immediately before the last session		↑ (posttest)
**TMS**								
Ben-Shachar et al., 1997 [83]	parallel	**Pulse freq.**: 25 Hz **Intensity:** 2.3 T **Pulse N:** 50 **Session N:** 1	healthy Sprague-Dawley rats	**Exp. group N:** 16–20 all male **Control group N:** 16–20 all male	brain tissue (frontal cortex)	10 s after the session	5-HT	ns (posttest)
							5-HIAA	ns (posttest)
							5-HIAA/5-HT	ns (posttest)
					brain tissue (hippocampus)		5-HT	↑ (posttest)
							5-HIAA	↑ (posttest)
							5-HIAA/5-HT	ns (posttest)
					brain tissue (striatum)		5-HT	ns (posttest)
							5-HIAA	ns (posttest)
							5-HIAA/5-HT	ns (posttest)
					brain tissue (midbrain)		5-HT	ns (posttest)
							5-HIAA	ns (posttest)
							5-HIAA/5-HT	ns (posttest)
Ben-Shachar et al., 1999 [84]	parallel	**Pulse freq.**: 15 Hz **Intensity:** 2 T **Pulse N:** 52 **Session N:** 10 **Session freq.:** 1/day	healthy Sprague-Dawley rats	**Exp. group N:** 18–20 all male **Control group N:** 18–20 all male	brain tissue (frontal cortex)	4 h after the last session	5-HT	ns (posttest)
					brain tissue (hippocampus)			ns (posttest)
					brain tissue (striatum)			ns (posttest)
					brain tissue (midbrain)			ns (posttest)
El Arfani et al., 2017 [85]	parallel	**Site:** prefrontal cortex **Pulse freq.**: 20 Hz **Intensity:** 110% MT **Pulse N:** 1560 **Session N:** 20 **Session freq.:** 5/day	healthy Sprague-Dawley rats	**Exp. group N:** 6 all male **Control group N:** 6 all male	brain tissue (striatum)	the day after the last session	5-HT	ns (posttest)
							5-HIAA	↓ (posttest)
Gur et al., 2000 [86]	parallel	**Pulse freq.**: 15 Hz **Intensity:** 1.12 T **Pulse N:** 600 **Session N:** 10 **Session freq.:** 1/day	healthy Albino rats	**Exp. group N:** 10 all male **Control group N:** 10 all male	microdialysis (left prefrontal cortex)	the morning after the last session	5-HT	ns (posttest)
Heath et al., 2018 [87]	parallel	**Site:** bifrontal cortex **Pulse freq.**: 10 Hz **Intensity:** 12 mT **Pulse N:** 1800 **Session N:** 20 **Session freq.:** 5/week	C57BL/6J mice with olfactory bulbectomy (modelling agitated depression)	**Exp. group N:** 11 all male **Control group N:** 9 all male	plasma	24 h after the last session	TRP	ns (posttest)
					plasma		5-HT	ns (posttest)
		**Intensity:** 90 mT		**Exp. group N:** 5 **Control group N:** 3	brain tissue (frontal cortex)		5-HT	ns (posttest)
				**Exp. group N:** 10 **Control group N:** 9	plasma		TRP	ns (posttest)
					plasma		5-HT	↓ (posttest)
		**Intensity:** 1.2 T		**Exp. group N:** 9 **Control group N:** 9	plasma		TRP	ns (posttest)
					plasma		5-HT	ns (posttest)
Kanno et al., 2003 [88]	parallel	**Site:** frontal brain **Pulse freq.**: 25 Hz **Intensity:** 0.6 T **Pulse N:** 125 **Session N:** 3 **Session freq.:** 1/day	healthy Wistar rats	**Exp. group N:** 8 all male **Control group N:** 7 all male	microdialysis (prefrontal cortex)	-	5-HT	ns (posttest)
Kanno et al., 2003 [89]	parallel	**Site:** frontal brain **Pulse freq.**: 25 Hz **Intensity:** 0.6 T **Pulse N:** 500 **Session N:** 1	healthy Wistar rats	**Exp. group N:** 6 all male **Control group N:** 6 all male	microdialysis (right prefrontal cortex)	during and 0–160 min after the session	5-HT	↓ (posttest)
Kanno et al., 2004 [90]	parallel	**Site:** frontal brain **Pulse freq.**: 25 Hz **Intensity:** 0.6 T **Pulse N:** 500 **Session N:** 1	healthy Wistar rats	**Exp. group N:** 6 all male **Control group N:** 6 all male	microdialysis (right dorsolateral striatum)	during and 0–160 min after the session	5-HT	ns (posttest)
Keck et al., 2000 [91]	parallel	**Site:** left frontal cortex **Pulse freq.**: 20 Hz **Intensity:** 130% MT **Pulse N:** 1000 **Session N:** 1	healthy Wistar rats	**Exp. group N:** 8 all male **Control group N:** 6 all male	microdialysis (right dorsal hippocampus)	during and 0–30 and 30–60 min after the session	5-HT	ns (group x time)
							5-HIAA	ns (group x time)
Kim et al., 2016 [92]	parallel	**Site:** right prefrontal cortex **Pulse freq.**: 10 Hz **Intensity:** 100% MT **Pulse N:** 1000 **Session N:** 15 **Session freq.:** 5/week	SHR/Izm rats (spontaneously hypertensive, modelling ADHD)	**Exp. group N:** 9 all male **Control group N:** 8 all male	brain tissue (prefrontal cortex)	7 h 50 min after the last session	5-HT	ns (posttest)
Löffler et al., 2012 [93]	parallel	**Site:** cerebral cortex **Pulse freq.**: 20 Hz **Intensity:** 130% MT **Pulse N:** 300 **Session N:** 1	healthy Wistar rats	**Exp. group N:** 8 all male **Control group N:** 6 all male	microdialysis (nucleus accumbens shell)	0–160 min after the session	5-HT	↑ (posttest/pretest)
							5-HIAA	ns (posttest/pretest)
Peng et al., 2018 [94]	parallel	**Site:** vertex **Pulse freq.**: 1 Hz **Intensity:** 0.84 T **Session N:** 7 **Session freq.:** 1/day	Sprague-Dawley rats with chronic unpredictable stress (modelling depression)	**Exp. group N:** 10 all male **Control group N:** 10 all male	brain tissue (prefrontal cortex)	24 h after the last session	5-HT	ns (posttest)
							5-HIAA	ns (posttest)
		**Intensity:** 1.26 T		**Exp. group N:** 10 **Control group N:** 10			5-HT	ns (posttest)
							5-HIAA	ns (posttest)
		**Pulse freq.**: 5 Hz **Intensity:** 0.84 T		**Exp. group N:** 10 **Control group N:** 10			5-HT	↑ (posttest)
							5-HIAA	↓ (posttest)
		**Intensity:** 1.26 T		**Exp. group N:** 10 **Control group N:** 10			5-HT	↑ (posttest)
							5-HIAA	↓ (posttest)
		**Pulse freq.**: 10 Hz **Intensity:** 0.84 T		**Exp. group N:** 10 **Control group N:** 10			5-HT	↑ (posttest)
							5-HIAA	↓ (posttest)
		**Intensity:** 1.26 T		**Exp. group N:** 10 **Control group N:** 10			5-HT	↑ (posttest)
							5-HIAA	↓ (posttest)
Poh et al., 2019 [95]	parallel	**Site:** lambda **Pulse freq.**: 10 Hz **Intensity:** 12 mT **Pulse N:** 6000 **Session N:** 1	healthy C57BL/6J mice	**Exp. group N:** 5 **M:F:** 60:40% **Control group N:** 5 **M:F:** 60:40%	brain tissue (cortex)	immediately after the session	5-HT	ns (posttest)
							5-HIAA	ns (posttest)
							5-HIAA/5-HT	↓ (posttest)
					brain tissue (hippocampus)		5-HT	ns (posttest)
							5-HIAA	ns (posttest)
							5-HIAA/5-HT	ns (posttest)
					brain tissue (striatum)		5-HT	ns (posttest)
							5-HIAA	ns (posttest)
							5-HIAA/5-HT	ns (posttest)
		**Intensity:** 1.2 T **Pulse N:** 3600		**Exp. group N:** 6 **M:F:** 50:50% **Control group N:** 5 **M:F:** 60:40%	brain tissue (cortex)		5-HT	ns (posttest)
							5-HIAA	ns (posttest)
							5-HIAA/5-HT	ns (posttest)
					brain tissue (hippocampus)		5-HT	ns (posttest)
							5-HIAA	ns (posttest)
							5-HIAA/5-HT	ns (posttest)
					brain tissue (striatum)		5-HT	ns (posttest)
							5-HIAA	ns (posttest)
							5-HIAA/5-HT	ns (posttest)
Wang et al., 2022 [96]	parallel	**Site:** center of skull **Pulse freq.**: 10 Hz **Intensity:** 90% MT **Pulse N:** 1000 **Session N:** 40 **Session freq.:** 5/week	Sprague-Dawley rats with spinal cord contusion injury	**Exp. group N:** 10 all female **Control group N:** 10 all female	ventral horn sections of the lumbar spinal cord	-	5-HT	ns (posttest)
**tDCS**								
Tanaka et al., 2013 [97]	parallel	**Site:** cortex **Polarity:** cathodal **Electrode size:** 5 mm × 5 mm **Intensity:** 800 μA **Duration:** 10 min **Session N:** 1	healthy Sprague-Dawley rats	**Exp. group N:** 7 all male **Control group N:** 7 all male	microdialysis (striatum)	during and 0–390 min after the session	5-HT	ns (posttest/pretest)
		**Polarity:** anodal		**Exp. group N:** 7 **Control group N:** 7				ns (posttest/pretest)

↑: Significantly increased or the experimental group levels were significantly larger than the control group; ↓: significantly decreased or the experimental group levels significantly smaller than the control group; ns: non-significantly changed or different; *: data not available for the featured group but taken from a larger group it belonged to or a subgroup of the majority of it; -: not applicable or data not available; N: count; freq.: frequency; exp.: experimental. For studies using the parallel design, “(group x time)” denotes the result based on group x time interaction, “(posttest)” group difference in the posttest, and “(posttest/pretest)” group difference in the posttest values standardized by the pretest ones. Here TRP refers to total TRP when unspecified.

**Table 4 ijms-23-09692-t004:** Outcomes of robust Bayesian meta-analysis.

Metabolite	Source	Studies	r	Effect BF10	Heterogeneity BF10	Publication Bias BF10
ECT						
TRP	plasma	N: 8 [35,39,43,48,52,53,56,58]	0.5	**0.216**	**10.191**	0.645
			0.7	**0.278**	**39.36**	0.891
			0.9	**0.183**	**>100**	**7.014**
KA	plasma	N: 2 [35,52]	0.5	**0.174**	0.568	0.656
			0.7	**0.174**	0.629	0.978
			0.9	**0.269**	2.267	1.015
KA/KYN	plasma	N: 2 [35,52]	0.5	**0.174**	0.468	0.461
			0.7	**0.178**	0.46	0.461
			0.9	**0.309**	0.782	0.517
**TMS**						
TRP	plasma	N: 2 [62,63]	0.5	**0.279**	0.699	0.633
			0.7	**0.242**	0.732	0.683
			0.9	**0.197**	1.306	1.271
5-HT	serum	N: 2 [62,63])	0.5	**5.46**	0.804	1.598
			0.7	**5.377**	0.777	1.678
			0.9	**9.232**	0.79	1.504

r: pretest-posttest correlation; BF10: inclusion Bayes factor, bolded when >3 or <1/3, suggesting moderate evidence for or against the presence of the meta-analytic item, respectively.

## Data Availability

No new data were created or analyzed in this study. Available data reviewed in this article is contained within the article or Appendix A.

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
