# Peer review of "Non-Invasive Brain Stimulation Effects on Biomarkers of Tryptophan Metabolism: A Scoping Review and Meta-Analysis"

_ijms, 2022, doi:10.3390/ijms23179692_

Round 1
Reviewer 1 Report
The paper is an excellent review contribution on the area of non invasive brain stimulation effect. The review is referred in particular on biomarkers of tryptophanmetabolism.
The paper reports with accurate details an extended set of contribution covering the various fields of the topic.
The authors have given a very useful contribution for people working in more areas of biomarkers for the study of the TRYPTOPHAN metabolism monitoring with non invasive techniques. The study is very impressive for the efforts in the synthesis of more parts.
The report could also be considered a strong example in making a survey.
The systematic method followed in arranging the scientific contributions is of excellent quality.
Moreover to furtherly improve the contribution it is suggested to include, from a bio enginerring poin of view some details and some references on the mathematical aspects on some biomedical signals. I suggest to include the following two contributions:
Proceedings of the Annual International Conference of the IEEE Engineering in Medicine and Biology Society, EMBSVolume 2015-November, Pages 4110 - 41134 November 2015 Article number 731929837th Annual International Conference of the IEEE Engineering in Medicine and Biology Society, EMBC 2015Milan25 August 2015 through 29 August 2015Code 116805
Document type
Conference Paper
Source type
Conference Proceedings
ISSN
1557170X
ISBN
978-142449271-8
DOI
10.1109/EMBC.2015.7319298
View more
Automatic preprocessing of EEG signals in long time scale
- Corradino C.Send mail to Corradino C.;
- Bucolo M.
Annual International Conference of the IEEE Engineering in Medicine and Biology-ProceedingsVolume 2, Pages 1981 - 19842001 Article number 261
Document type
Article
Source type
Journal
ISSN
05891019
DOI
10.1109/IEMBS.2001.1020618
View more
Independent component analysis of magnetoencephalography data
- Fortuna L.a;
- Bucolo M.a;
- Frasca M.a;
- La Rosa M.a;
- Shannahoff-Khalsa D.S.b;
- Schult R.L.b;
- Wright, Jon A.b
Moreover what is the contribution of noise (electrical or biochemical ) in the context of this study ? Brain noise inspires some important research. It will be furtherly important to cite this effects and to take into account of the following contribution.
Journal of the Physical Society of JapanVolume 90, Issue 71 July 2021 Article number 075002
Document type
Review
Source type
Journal
ISSN
00319015
DOI
10.7566/JPSJ.90.075002
View more
Can noise in the feedback improve the performance of a control system?
- Bucolo, Maidea, b;
- Buscarino, Arturoa, bSend mail to Buscarino A.;
- Fortuna, Luigia, b;
- Gagliano, Salvinaa
Author Response
Response to Reviewer 1
Moderate English changes required
Response: Thank you for your suggestion – a native-English speaking co-author has proofread the manuscript. Revisions are tracked in our updated submission.
The paper is an excellent review contribution on the area of non invasive brain stimulation effect. The review is referred in particular on biomarkers of tryptophan
metabolism.
The paper reports with accurate details an extended set of contribution covering the various fields of the topic.
The authors have given a very useful contribution for people working in more areas of biomarkers for the study of the TRYPTOPHAN metabolism monitoring with non invasive techniques. The study is very impressive for the efforts in the synthesis of more parts.
The report could also be considered a strong example in making a survey.
The systematic method followed in arranging the scientific contributions is of excellent quality.
Response: Thank you for your kind words!
Moreover to furtherly improve the contribution it is suggested to include, from a bio enginerring poin of view some details and some references on the mathematical aspects on some biomedical signals. I suggest to include the following two contributions:
[Corradino & Bucolo (2015). Automatic preprocessing of EEG signals in long time scale. doi: 10.1109/EMBC.2015.7319298.]
[Fortuna et al. (2001). Independent component analysis of magnetoencephalography data. doi: 10.1109/IEMBS.2001.1020618.]
Moreover what is the contribution of noise (electrical or biochemical ) in the context of this study ? Brain noise inspires some important research. It will be furtherly important to cite this effects and to take into account of the following contribution.
[Bucolo et al. (2021). Can Noise in the Feedback Improve the Performance of a Control System? doi: 10.7566/JPSJ.90.075002]
Response: Thank you, and we agree, understanding neural signal analysis (e.g., obtained by EEG and MEG) can inform the relationship between therapeutic NIBS effects on tryptophan metabolism, e.g., whether neural activity modulations are better predictors of changes to biomarker levels. However, we were advised by the editors to focus our discussion on molecular content. As such, and after carefully readings these suggested works, we cited them as essential techniques to use in future experiments on our review research questions.
“For instance, whether change in brain activity and noise following successful therapeutic NIBS coincides to changes in biomarker levels is an important research question to be investigated with careful signal analysis approaches, such as [143-145]”.
Reviewer 2 Report
It is an interesting systematic review which explored the published work on non-invasive brain stimulation effects on biomarkers of tryptophan metabolism for the past 50 years.
1. Proofreading should be conducted.
2. Of the included 65 studies, only two tDCS studies are included. Is it due to the non-availability of the full publication?
Author Response
Response to Reviewer 2
English language and style are fine/minor spell check required
- Proofreading should be conducted.
Response: Thank you for your suggestion – a native-English speaking co-author has proofread the manuscript. Revisions are tracked in our updated submission.
- Of the included 65 studies, only two tDCS studies are included. Is it due to the non-availability of the full publication?
Response: Thank you for your question. The low count of tDCS studies is not due to unavailable full publications. Based on title and abstracts of the two studies we could not retrieve, one was a TMS preclinical study (Mano et al., 1989; Table S2) and the other was an ECT clinical study (Abrams et al., 1976; Table S2).
